# Spatio-Temporal Changes and Habitats of Rare and Endangered Species in Yunnan Province Based on MaxEnt Model

**Yiwei Lian** [1,2,3], **Yang Bai** [1,2,3,*], **Zhongde Huang** [1,2,3], **Maroof Ali** [1], **Jie Wang** [1,2,3] and **Haoran Chen** [4]

1   Center for Integrative Conservation & Yunnan Key Laboratory for Conservation of Tropical Rainforests and Asian Elephants, Xishuangbanna Tropical Botanical Garden, Chinese Academy of Sciences, Mengla 666303, China; lianyiwei@xtbg.ac.cn (Y.L.); huangzhongde@xtbg.ac.cn (Z.H.); maroof@xtbg.ac.cn (M.A.); wangjie2021@xtbg.ac.cn (J.W.)
2   Yunnan International Joint Laboratory of Southeast Asia Biodiversity Conservation, Jinghong 666303, China
3   University of Chinese Academy of Sciences, Beijing 100049, China
4   Institute of International Rivers and Ecological Security, Yunnan University, Kunming 650500, China; chenhaoran@xtbg.ac.cn
*   Correspondence: baiyang@xtbg.ac.cn

**Abstract:** Biodiversity is crucial for ecosystem functioning, but it is rapidly declining due to human activities and climate change. Protecting biodiversity has become a key priority for global environmental conservation actions. Rare and endangered species have a great impact on the ecosystem, yet due to their limited survival capacity, they are more prone to extinction, thus exerting a significant impact on biodiversity. However, current research reveals a lack of information concerning the potential distribution and changes of these species. This study used the maximum entropy model to predict the present and future potential habitats of rare and endangered species in Yunnan Province. After superimposing model results, four richness regions are divided by the natural breakpoint method and analyzed. Existing protected areas are compared with hotspots, and the land-use composition of hotspots is also analyzed. The results revealed that, in both current and future scenarios, rare and endangered species in Yunnan Province are primarily found in the western mountainous region, the Xishuangbanna–Wenshan high temperature area, and the Kunming–Qujing dense vegetation cover area. These species are also expanding their distribution towards the western mountainous area. However, under the low carbon emission scenario (RCP2.6), these species will spread from the high abundance regions to the low altitude hotspots by 2070. In the high carbon emissions scenario (RCP8.5), there will be fewer high abundance areas in 2070 than in 2050. The transfer matrix analysis reveals regional richness variations over time. Furthermore, the analysis revealed significant conservation gaps and found that existing hotspot areas were heavily affected by human activities. To improve conservation efficiency, it is necessary to enhance the protection of existing hotspots in Yunnan Province. Climate change plays a significant role in species migration, with precipitation levels being a key factor. The necessary actions should be taken to address the insufficient protection, resolve conflicts between human activities and land use in critical areas, and formulate effective strategies for adapting to future climate changes. Yunnan Province, with its rich species resources, has the potential to become a global innovator in biodiversity conservation by implementing improved conservation strategies.

**Keywords:** rare and endangered species; MaxEnt model; biodiversity prediction; climate change; gap analysis

## 1. Introduction

With global environmental changes and explosive population growth, various factors, such as habitat loss, habitat fragmentation, resource overuse, invasion of alien species, environmental pollution, and human-induced climate change, along with barriers to biological reproduction, have led to the degradation of ecosystems and loss of biodiversity [1].

Biodiversity loss is a significant global problem that worsens over time [2]. Currently, biodiversity faces five major pressures: climate change, habitat loss and degradation, nutrient overenrichment and pollution, overexploitation and unsustainable use, and invasive alien species [3]. Furthermore, a conflict arises between economic level and national consumption level development and biodiversity conservation [4]. The monitoring of endangered species by the relevant authorities is inadequately implemented [5]. The overall trend of biodiversity loss has not been effectively controlled, and China continues to cope with a multitude of issues stemming from its economic development, which in turn puts a significant strain on biodiversity [6].

Studies have demonstrated the significant impact of losing various life forms on the structure and function of entire ecosystems. It also affects various ecosystem services [7,8]. This is particularly concerning for rare and endangered species (RESs) which have low abundances and are more vulnerable to environmental change and extinction [9–11]. These species often have small and fragmented geographical ranges [12]. These characteristics correspond to species on the IUCN Red List of species on the six threatened levels of near threatened, vulnerable, endangered, critically endangered, extinct in the wild, and extinct. The contribution of RESs to ecosystems is crucial [13]. Furthermore, RESs have a poor ability to reproduce and spread [14] and will become extinct if they are not protected [15], which has greater significance for biodiversity. Biodiversity can have catastrophic consequences for ecosystems, as it is crucial for human health and well-being. Therefore, reducing biodiversity loss and protecting animal and plant resources are included in the Millennium Development Goals and Action Goal Three of the Kunming–Montreal Global Biodiversity Framework (GBF) [16,17].

At present, the Earth's climate is experiencing rapid dynamic change [18]. These changes are negatively impacting the habitats of RESs, leading to a decline in biodiversity indicators [19]. Despite this, biodiversity pressure remains high, creating a correlation between biodiversity change and climate change. While most organisms have some degree of adaptability to environmental changes, human activity has reduced their ability to adapt [20]. Therefore, predicting potential hotspots for RESs can help conservation managers understand how species distributions may change under these circumstances.

Of the many ways to conserve biodiversity, in situ conservation, particularly through the establishment of protected areas (PAs), is the most effective method for conserving biodiversity [21,22]. PAs are cost efficient and play a crucial role in addressing biodiversity loss [23–25], and establishing protection systems is key to all effective biodiversity conservation tools [26]. However, even with the existence of PAs, species populations both inside and outside PAs continue to decline [27]. Many PAs established before 1992 are facing increasing human pressure, and changing environmental conditions may compromise their ability to protect species and ecosystems in the future [28]. Therefore, it is urgent to optimize the distribution pattern of PAs to effectively protect RESs.

The maximum entropy (MaxEnt) model is a very effective model for predicting a species' geographical distribution. It can only use the existing data to run the model, and the prediction accuracy is high, which avoids data overfitting [29–31]. The MaxEnt model can achieve the maximum level of randomness in a generic context, without being restricted by pattern limitations [32]. The Maxent model's predictions are closer to realistic niches [33], so after the model passes the validation threshold, the results are highly available. The MaxEnt model has been widely used to predict the habitat distribution of RESs, and Chinese scholars have used it to predict the distribution area of *Cornus officinalis* and *Thuja sutchuenensis Franch.* [34,35]. Additionally, it has been used to study the geographical ranges of endangered species around the world, like *Ctenomys magellanicus*, *Cryptobranchus alleganiensis*, and wild *Nepeta crispa* [36–38]. The forecast results were excellent. Previous studies used different methods to identify conservation priority areas in Yunnan Province [39–41]. Some of these studies have applied the MaxEnt model to explore the distribution of RESs in Yunnan Province [42–44]. However, there is still a lack of research that applies the MaxEnt model to predict priority PAs in Yunnan Province. This study aims to fill the research gap

by considering species distribution patterns under future environmental changes. The future environmental prediction model relies on four representative concentration pathways (RCPs) that depict various emission trajectories. These RCPs include emission substances, emission concentrations, and land-use trajectory. As the carbon dioxide concentration in the atmosphere increases, the Earth's temperature rises. The RCPs are named after the radiative forcing target levels for 2100, which are estimated based on emissions and other influencing agents. Four selected RCPs were considered in this study: RCP2.6 (very low forcing levels), RCP4.5/RCP6 (moderately stable scenarios), and RCP8.5 (very high baseline emission scenario).

This study assessed the efficacy of existing PAs by comparing the distribution of simulated hotspots of RESs with that of existing PAs. The MaxEnt model was used to predict the changes in the potential distribution area of RESs under future climate-change scenarios (2050, 2070). The study extracted land-use data corresponding to high species-rich regions. It analyzed the impact of major human activities on the potential distribution of RESs. In this study, ArcGIS software was used to generate a habitat-change transfer matrix, enabling the calculation of future increases or decreases in potential distribution areas. This approach provides an effective means to quantify the potential habitat changes of vegetation influenced by climate change.

This study aims to address the following questions: (1) are the existing PAs in Yunnan Province effectively protecting the potential habitats of RESs? (2) Which land-use types have the most significant impact on the potential habitats of RESs? What are the main land-use types in the hotspot area? (3) How will the potential distribution of RESs change under different emission patterns (RCP2.6, RCP4.5, RCP6.0, and RCP8.5) under future climate change (2050, 2070)?

## 2. Materials and Methods

### 2.1. Study Area

The study area is in Yunnan Province, located between 97°31′~106°11′ east longitude and 21°8′~29°15′ north latitude (Figure 1). The province is adjacent to Guangxi Zhuang Autonomous Region and Guizhou Province in the east, separated from Sichuan Province by the Jinsha River in the north, close to Xizang Autonomous Region in the northwest, adjacent to Myanmar in the west, and bordering Laos and Vietnam in the south and southeast, respectively. Yunnan Province, along with Guangxi Zhuang Autonomous Region and Hainan Island, form the Indo–Burma region in the Southeast Asian biodiversity hotspot [45], which is of paramount significance for global biodiversity conservation.

Yunnan Province has a total area of 394,100 km$^2$, accounting for 4.1% of the country's total area. It is characterized by five major landforms: mountains, hills, basins, plateaus, and small plains. The diverse regional terrain is the primary reason for the conservation of numerous species in Yunnan Province. Additionally, the province has a rich climate belt and a variety of climates. Yunnan Province has a cold, warm, and hot (including subtropical) climate. Most of Yunnan belongs to a subtropical plateau monsoon climate, and Northwest Yunnan belongs to a cold zone climate. Eastern and Central Yunnan have a temperate climate. The south and southwest of Yunnan Province belong to the tropics, showing the climate characteristics of dry and hot valleys [46]. Therefore, Yunnan creates a habitat suitable for the growth of many species. As of 2016, a total of 2253 species of seed plants belonging to 225 families and 2140 genera have been identified in Yunnan, making it the province with the highest number of plant species in China [47].

According to the Guiding Opinions on Establishing a System of Protected Natural Areas with National Parks as the main body [48], China's protected natural areas, which are primarily composed of national parks, can be categorized into three categories: national parks, nature reserves (including national, provincial, municipal and county levels), and natural parks. This study focuses on five types of natural parks: wetland parks, forest parks, scenic areas, parks, and stone desert parks [49]. The distribution of these protected-area types is shown in Figure 1.

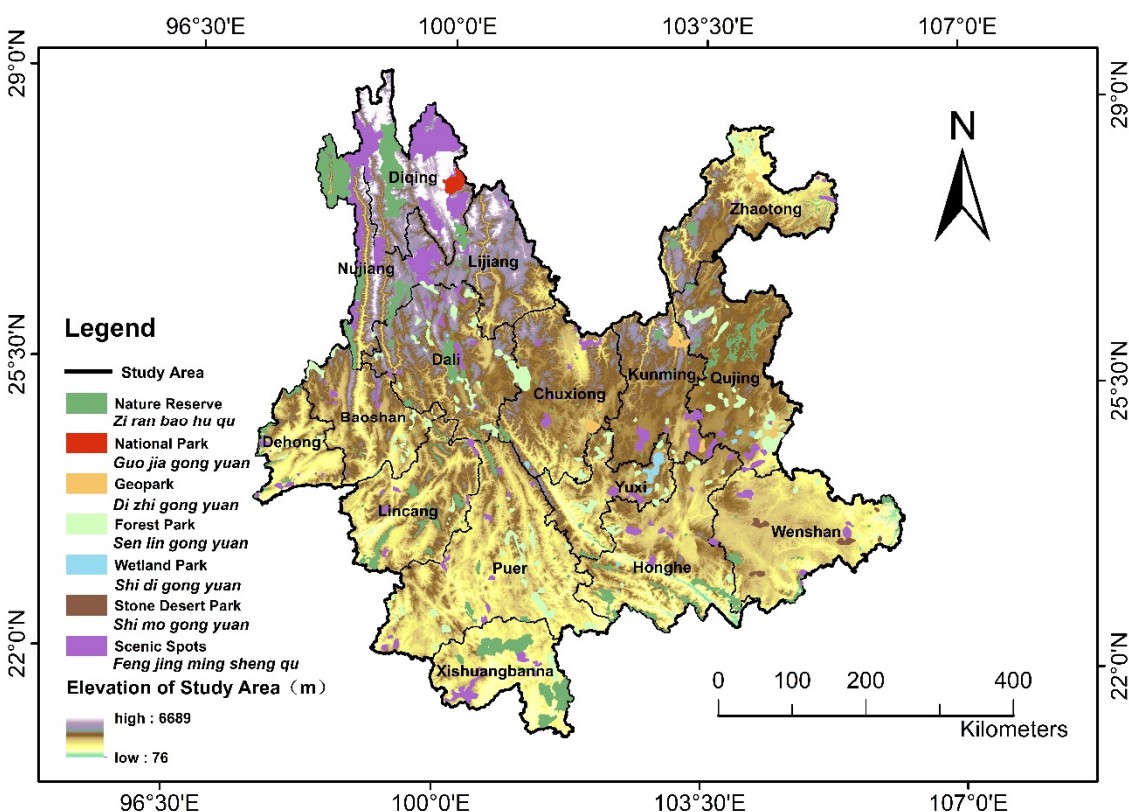

**Figure 1.** Spatial distribution of PAs in the study area, with terrain and landforms.

### 2.2. Species Occurrence Data

This study focused on the conservation of vulnerable, near threatened, endangered, critically endangered, extinct, and extinct wild species in the study area. The classification of endangered species is based on Table S1. The selection criteria for these species were based on the IUCN Red List of Threatened Species [50]. Species distribution data were obtained from the IUCN Red List of Threatened Species and the Global Biodiversity Information Facility [51]. After downloading the data package from the website, a massive amount of data was screened to select plant records with clear spatial distribution information, while removing erroneous and duplicate records. The taxonomic names were cross-checked with the latest valid species names in the Catalogue of Life [52]. Plant-acceptable names have been compared with the World Flora online database [53]. A total of 604 species from 164 families and species groups, along with 3926 distribution points for representative RESs, were collected in this research (Table S2).

### 2.3. Environment Variables

The specific sources of the environmental elements used in this study are shown in Table S3. The digital elevation model (DEM) data was sourced from the Geospatial Data Cloud [54], and 19 biological climate variables were sourced from the WorldClim 2.1 version [55], representing average values from 1970 to 2000, with a spatial resolution of 30 seconds/km$^2$. Climate-prediction data were obtained from WorldClim, based on BCC-CSM1-1 [56]. (Table S3). Data for four emission pathways (RCP2.6, RCP4.5, RCP6.0, and RCP8.5) and two time points (2050 and 2070) were downloaded. By 2100, the rising radiative forcing pathway led to ~3 W/m$^2$, 4.5 W/m$^2$, 6.0 W/m$^2$, and 8.5 W/m$^2$ [57,58]. To avoid overfitting, a correlation analysis of 19 bioclimatic variables was carried out (Table S4). Finally, 10 variables were selected, and a total of 14 variables, including elevation, slope, slope direction, and normalized difference vegetation index (NDVI) [59], were input into the model. The downloaded data were preprocessed with the relevant tools of ArcGIS10.2. The 14 biological climate variable layers are extracted from Yunnan Province using the mask

extraction tool of ArcGIS10.2, and, then, random points are created for these extracted layers. Interpolation (Kriging interpolation) is performed using random points, and, finally, the interpolated study-area range is converted to an ASCII file for saving.

### 2.4. Distribution Modeling

Using ArcGIS 10.2 as the platform, environmental variable raster files were filtered for specific variables. In addition, the coordinate system, boundary, and resolution of all layers were unified. To avoid low AUC values caused by the large research area or small distribution points, species data were imported into the MaxEnt model by families.

In order to ensure the model's normal operation and performance, species with loci $\geq 6$ were imported as a whole. Species with loci less than 6 loci were integrated based on their distribution characteristics. The integrated species were categorized into three groups: animals, plants, and fungi.

In this study, 75% of the species distribution information was used to build the model, while 25% was used for model verification. The background points were set to 10,000 and heterogeneous threshold parameters were used to transform the model output results. The other parameters were set to the model default values [42].

This research evaluates the prediction accuracy of the MaxEnt model using the area under the receiver operating characteristic (ROC) curve (area under curve, AUC). The evaluation of AUC values generally adheres to the following standards: $0.9 < AUC < 1.0$, the model prediction result is extremely accurate; $0.7 < AUC < 0.8$, the prediction accuracy is high; $0.6 < AUC < 0.7$, the prediction accuracy is low; and $0.5 < AUC$, the model prediction fails [60]. For model verification and testing, 25% of the species distribution points were selected; when $AUC \geq 0.7$, the simulated species potential habitat of the model was used for further analysis for biodiversity conservation planning. An AUC greater than 0.75 was then considered acceptable. To create a biodiversity hotspot distribution map, the species distribution probability map was transformed into a floating-type raster using ArcGIS tools. The distribution probability map was then converted into a hotspot distribution map based on the minimum training threshold.

### 2.5. Model Accuracy and Validation

The model used to simulate the distribution of RESs in Yunnan Province is acceptable for both current and future scenarios (RCP2.6, RCP4.5, RCP6.0, RCP8.5, 2050, and 2070). The average AUC values for RESs in all scenarios were above 0.75 (Table S5). This indicates the success of the model in evaluating species distribution within the study area. The specific AUC values are listed in the Supplementary Materials) in this article.

### 2.6. Gap Analysis

The protection gap in this study refers to the insufficient space available for RESs in the existing protected-area system. The identified priority protection hotspots and the established protected areas were superimposed on the space analysis. Those space areas without overlap are protection gap areas. The results of this gap analysis method in previous studies have substantial implications for species conservation [61,62].

In order to evaluate the richness of the potential habitat of creatures identified by the model, creature hotspots were identified. The results of the model operation are binarized according to the 10% training threshold generated by the model. The part larger than the training threshold is selected as the potential distribution area of the family. The layers of each family/species group after binarization are overlaid by a raster calculator, and the layers are divided into four levels using the natural breakpoint method [63], in order of richness from high to low. And then, count the area and proportion of each land use and land cover in the hotspot areas. This study obtained land-use and land-cover data. The data set is provided by the Data Center for Resources and Environmental Sciences, Chinese Academy of Sciences (RESDC) [64]. Then, based on the area and proportion of hotspots,

our research determined the natural community types that RESs rely on. It also identified the human development that poses the greatest threat to potential habitats.

In order to evaluate the protection effect of existing reserves on representative creature communities in Yunnan Province, the study examined the simulated hotspots with the existing reserves. Then, extract the hotspots inside and outside the protected area, calculate their area and proportion, and evaluate the protection efficiency of the current protected area. Finally, this study selected regions with a predicted abundance greater than 25% of RESs from the simulation results as the priority protection area.

In addition, this research also uses data on RESs downloaded from IUCN to build the fishing net tool to evaluate the number of RESs in each grid memory. Visualize this quantity of data on a map to get the result. Using this result, we compare the existing protection of the RESs in IUCN and the MaxEnt model to predict the distribution of RESs.

### 2.7. Predicting Potential Habitat under Future Climate Change

In order to evaluate the impact of future climate change on creature communities, this study used the MaxEnt model to predict the potential habitats of RESs in Yunnan Province in 2050 and 2070 under four emission modes: RCP2.6, RCP4.5, RCP6.0, and RCP8.5. In addition to comparing the hotspot distributions over time, as well as the changing area and spatial distribution, the results were plotted on a binary graph. In order to identify future protection gaps under predicted climate change, hotspots simulated for 2050 and 2070 were compared with current hotspots for RESs. Finally, by extracting the area of each richness and the overlap between them, the transfer of richness levels over time scales is obtained. Using these areas, we created a transfer matrix to quantify the changes in RES hotspots from current to future points. In this study, the vector area was calculated by the ArcGIS field calculator, and the raster area was calculated by grid number $\times$ grid size (900 m$^2$).

For the purpose of analyzing the changing trend of the high-richness region in the future, 360 radiation lines are made in the average center of the study area, and the overlap length of each radiation line with the high-richness region in this direction is taken. The cross length in each direction was used to qualitatively indicate how much or how less of a high-richness area exists in that direction. This approach can show the direction and amount of high-richness area transfer under each emission scenario. The impact factor contribution ranking is generated when the model is run. The number of times that each factor appeared in the top five contribution rates was counted to find out the driving factors with higher contribution rates, and the reasons for the changes in each region under climate change were analyzed through the driving factors.

## 3. Results

### 3.1. Current Habitats of RESs

The rich distribution area for endangered species comprises 80.54% of Yunnan Province (Figure 2). Species distribution in high richness areas (HRA, richness < 30.84 were concentrated in the northwest and east of Yunnan Province and distributed in small areas in Xishuangbanna. This was with a total area of 25,080.80 km$^2$. As regards the number of RESs in the grid, Yunnan contained an 'unsuitable' area (UA, richness < 7.90) of 1,171,531.15 km$^2$, 'low richness' area (LRA, richness < 17.84) of 121,575.17 km$^2$, and 'medium richness' area (MRA, richness < 30.84) of 66,266.61 km$^2$. Almost half of the UA showed a concentrated distribution in the central and the north areas of Yunnan (Figure 2).

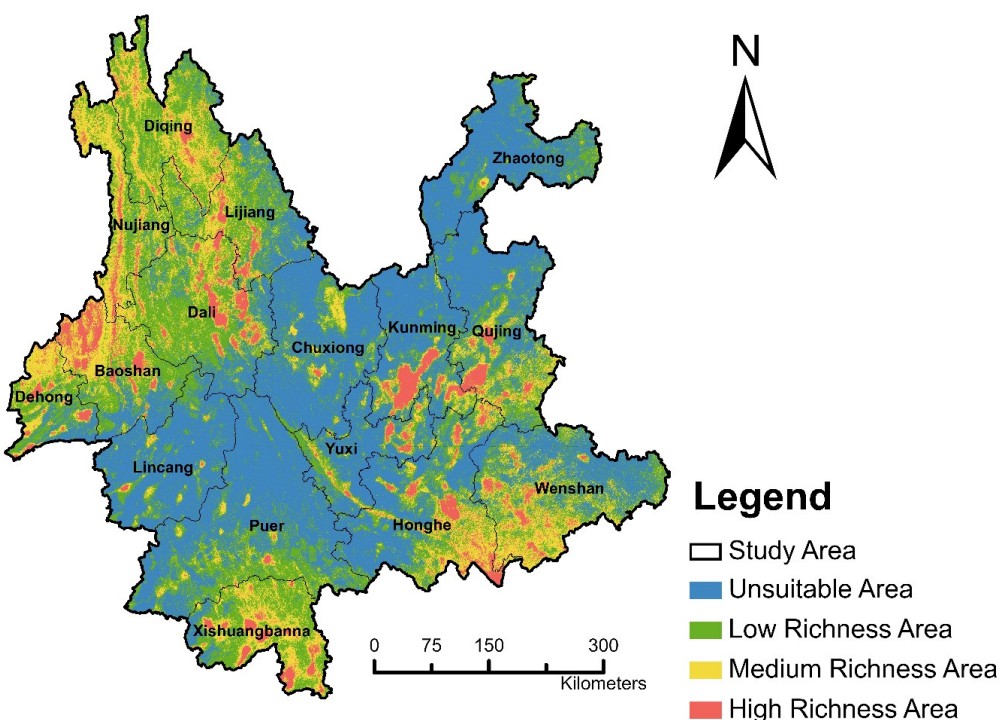

**Figure 2.** Potential distribution map after stacking analysis of all species.

*3.2. Future Habitats of RESs*

In the future scenario, the concentrated distribution area of RESs is roughly unchanged (Figure 3). The spatial change flow in the figure starts at the current distribution center. The changes in areas with high abundance in different scenarios and years are shown in Figure S1.

The distribution area of the high species richness zone was calculated under four different scenarios. The results showed that in 2050 the largest hotspot area was observed under the RCP4.5 emission scenario. In contrast, the smallest hotspot area was observed under the RCP6.0 emission scenario. In the year 2070, the same remains accurate. Among the four scenarios, only the RCP8.5 scenario decreases hotspot areas between 2050 and 2070. In contrast, the other scenarios expand the area of hotspots in the period from 2050 to 2070 to a certain extent (Figure 3).

To represent the distribution of hotspot abundance in different directions, the intersection length of hotspots in Yunnan Province was calculated under four scenarios (Figure 4). The results revealed that, under the RCP2.6 emission scenario, the hotspot distribution in 2070 increased significantly towards the west and south. Under the RCP4.5 emission scenario, the hotspots towards the west increased in both 2050 and 2070, while the hotspots towards the south and east increased substantially in 2050. In the RCP6.0 emission scenario, there was not much difference in the main distribution directions across the three time periods. However, distribution areas in the east and west significantly increased in future scenarios. Within the RCP8.5 emission scenario, the geographical distribution of hotspots will shift from the southeastern region to the eastern region in the future. The extent of hotspots in the western region will experience a significant increase by 2050.

The model calculates the contribution of bioclimatic variables to species distribution. The contribution rate of each type of factor is calculated comprehensively. The contribution rate of the precipitation factor was 31.96%, the contribution rate of the normalized vegetation index was 16.52%, and the contribution rate of the topographic factor was 14.57%. The most important contributing factors are annual precipitation, normalized vegetation index, and slope.

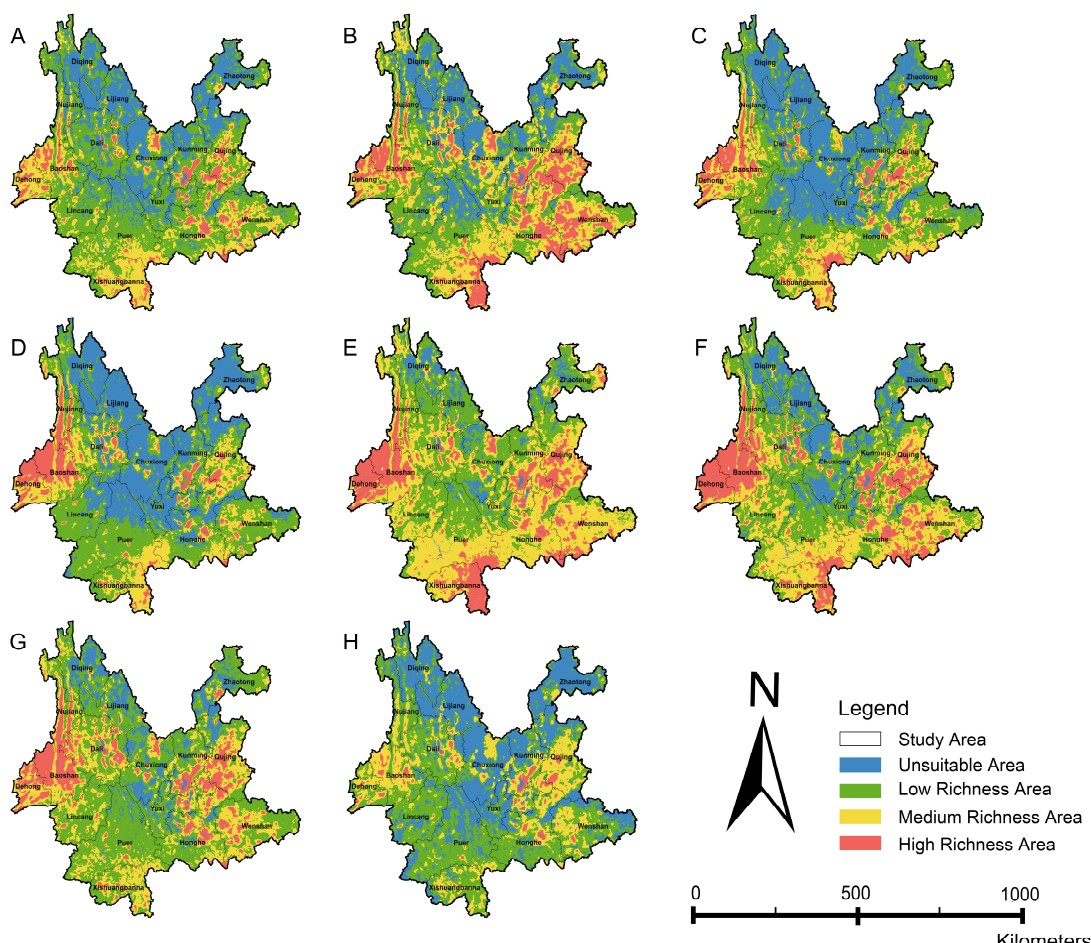

**Figure 3.** The predicted distribution of RESs in the future environment. (**A–H**) represents 8 different emission models, respectively. (**A**): RCP2.6, 2050; (**B**): RCP4.5, 2050; (**C**): RCP6.0, 2050; (**D**): RCP8.5, 2050; (**E**): RCP2.6, 2070; (**F**): RCP4.5, 2070; (**G**): RCP6.0, 2070; (**H**): RCP8.5, 2070).

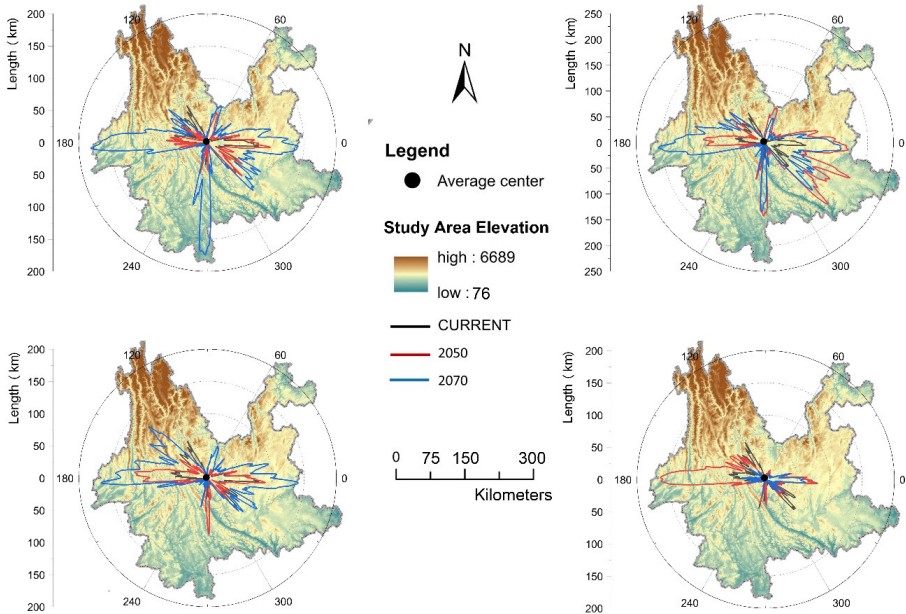

**Figure 4.** Comparison of the distribution of HRAs from different angles in four scenarios (RCP2.6, RCP4.5, RCP6.0, and RCP8.5).



### 3.3. Regional Transfer Matrix of Different Richness Levels

Future climate change will have a significant impact on the potential habitats of ecologically significant species. Under the emission mode of RCP2.6, by 2050, only 45.81% of potential habitats in LRAs will be retained, approximately 23.20% will become UAs, 40.67% of potential habitats in MRAs will be retained, 11.67% will become UAs, 53.22% of potential habitats in HRAs will be conserved, and 1.41% of high adaptation is transformed into UA (Table 1). The abundance area transfer matrix under the RCP4.5, RCP6.0, and RCP8.5 scenarios is in Table S6.

**Table 1.** The abundance area transfer matrix under RCP2.6 scenarios.

| | | UA | LRA | MRA | HRA | Total (km$^2$) |
|---|---|---|---|---|---|---|
| | | | Rcp2.6 2050 | | | |
| | UA | 59,808.11 | 94,941.85 | 16,464.49 | 316.7 | 171,531.15 |
| | LRA | 28,206.38 | 55,687.57 | 34,874.07 | 2807.15 | 121,575.17 |
| 2022 | MRA | 7730.65 | 23,720.17 | 26,950.82 | 7864.97 | 66,266.61 |
| | HRA | 354.54 | 3856.64 | 7520.64 | 13,348.98 | 25,080.8 |
| | Total (km$^2$) | 96,099.68 | 178,206.23 | 85,810.03 | 24337.8 | 384,453.74 |
| | | | Rcp2.6 2070 | | | |
| | | UA | LRA | MRA | HRA | Total (km$^2$) |
| | UA | 26,126.3 | 67,924.49 | 2043.84 | 5.00 | 96,099.63 |
| Rcp2.6 | LRA | 336.95 | 63,359.52 | 111,492.86 | 3016.89 | 178,206.23 |
| 2050 | MRA | 85,810.03 | 501.88 | 44,774.67 | 40,533.48 | 171,620.06 |
| | HRA | 24,337.8 | 24,337.8 | 545 | 23,760.05 | 72,980.65 |
| | Total (km$^2$) | 136,611.07 | 156,123.69 | 158,856.38 | 67,315.43 | 518,906.57 |

Under future climate change, these areas will undergo significant changes that will affect the habitats and ecology of the protected areas. Under the emission mode of RCP2.6, from 2050 to 2070, only 35.55% of potential habitats in LRA will be retained, approximately 0.19% will become UAs, 26.09% of potential habitats in MRA will be retained, 50.00% will become UAs, 32.56% of potential habitats in HRAs will be protected, and 33.35% of HRA is transformed into UAs.

The proportion of the shift in richness for every emission mode is shown in Table 1. Period 1 refers to 2022~2050, and period 2 refers to 2050~2070. It can be found that what happens in the near time scale is mostly the transition from the LRA to the UA, while what happens in the distant time scale is mostly the transition from the HRA to the UA (Table 2 and Table S6).

**Table 2.** On time scale, the proportion of each richness area retained and turned into UAs.

| Emission Mode | Period | Retention Ratio of LRA | The Proportion of LRA to UA | Retention Ratio of MRA | The Proportion of the MRA to UA | Retention Ratio of HRA | The Proportion of the HRA to the UA |
|---|---|---|---|---|---|---|---|
| RCP2.6 | 1 | 45.81% | 23.20% | 40.67% | 11.67% | 53.22% | 1.41% |
| | 2 | 35.55% | 0.19% | 26.09% | 50.00% | 32.56% | 33.35% |
| RCP4.5 | 1 | 30.47% | 18.88% | 37.20% | 9.27% | 74.14% | 1.11% |
| | 2 | 68.16% | 7.14% | 64.40% | 0.97% | 39.68% | 50.01% |
| RCP6.0 | 1 | 43.06% | 26.71% | 39.90% | 14.24% | 56.32% | 2.70% |
| | 2 | 59.53% | 0.92% | 52.03% | 0.04% | 41.41% | 50.00% |
| RCP8.5 | 1 | 44.57% | 25.96% | 36.31% | 12.90% | 57.55% | 2.85% |
| | 2 | 33.86% | 9.44% | 51.75% | 1.92% | 26.03% | 0.00% |

### 3.4. Gap Analysis and Priority Conservation Identification

Overlapping the existing PAs with the current HRA, it was found that the overlapping area of the two is 1369.36 km$^2$, accounting for 5.46% of the total HRA of 23,017.98 km$^2$, and

thus for 80.66% (Table 3). If the measured area is replaced by the current hotspot area, then the overlap area of PAs and hotspot area is 18,075.44 km², accounting for 19.34% of the total hotspot area of 93,471.13 km², representing 27.57% of the PAs area, and accounting for 4.59% of the area of Yunnan Province (Figure 5).

**Table 3.** The protection ratio of existing PAs to HRA and the protection efficiency of PAs under different scenarios.

| Emission Mode | HRA (km²) | Overlapping Area (km²) | UA (km²) | Protection Ratio | Unprotected Proportions | Proportion of Effective PAs |
|---|---|---|---|---|---|---|
| Current | 25,080.80 | 1369.36 | 23,017.98 | 5.46% | 94.54% | 6.23% |
| RCP2.6, 2050 | 24,337.80 | 919.63 | 22,093.26 | 3.78% | 96.22% | 4.18% |
| RCP2.6, 2070 | 67,315.43 | 3828.49 | 65,058.48 | 5.69% | 94.31% | 17.42% |
| RCP4.5, 2050 | 61,505.38 | 2826.70 | 59,111.00 | 4.60% | 95.40% | 12.86% |
| RCP4.5, 2070 | 69,112.58 | 2891.65 | 66,756.81 | 4.18% | 95.82% | 13.15% |
| RCP6.0, 2050 | 29,774.14 | 914.91 | 27,245.38 | 3.07% | 96.93% | 4.16% |
| RCP6.0, 2070 | 49,940.75 | 1281.91 | 47,524.70 | 2.57% | 97.43% | 5.83% |
| RCP8.5, 2050 | 32,591.40 | 1092.00 | 30,149.49 | 3.35% | 96.65% | 4.97% |
| RCP8.5, 2070 | 8697.00 | 145.30 | 5717.27 | 1.67% | 98.33% | 0.66% |

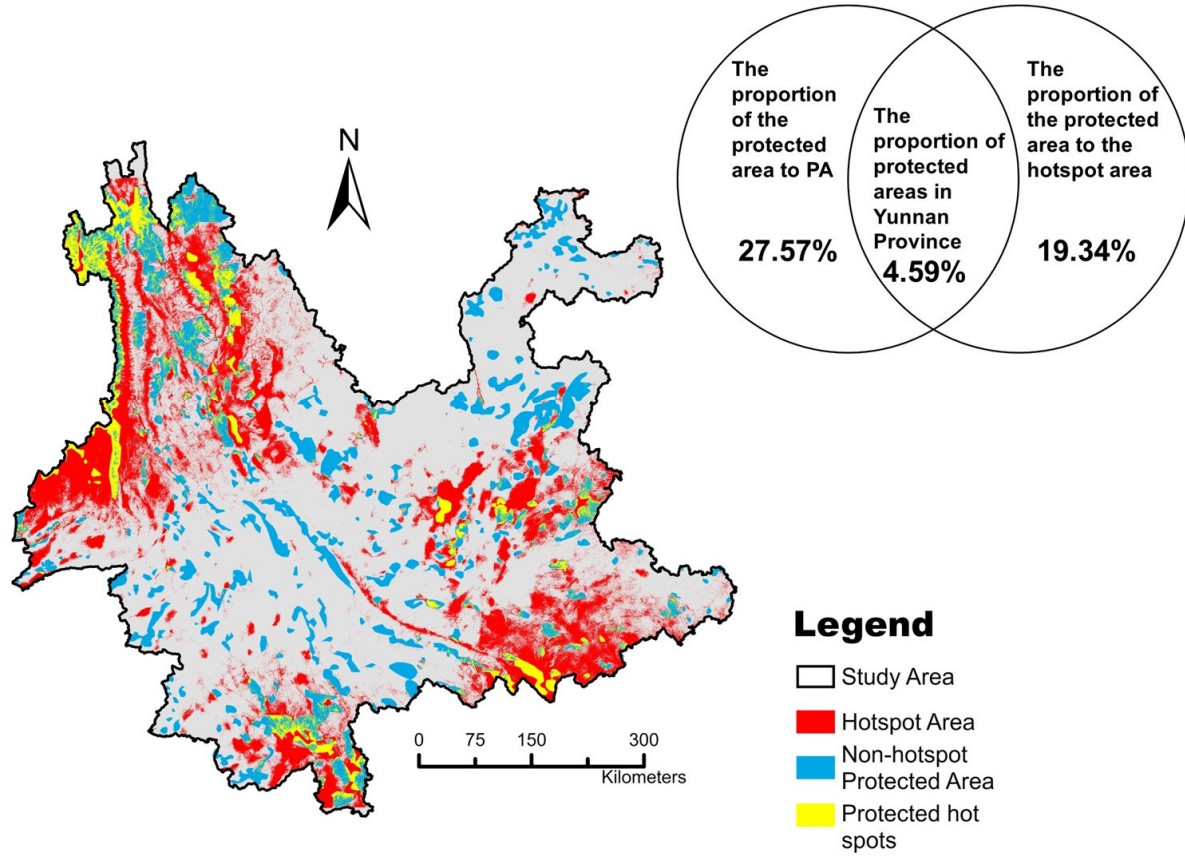

**Figure 5.** Assessing the distribution pattern of the RESs hotspots and PAs.

In Table 3, the protection ratio of the existing PAs to the selected species' HRA in the future scenario is listed. The protection efficiency of the PAs under this scenario is represented by the proportion of the existing PAs containing hotspots.

The IUCN data prediction results are divided into hotspots and nonhotspot areas according to the high richness threshold 31 of the MaxEnt model. There is a big difference between the hotspots of the MaxEnt model (Figure S2), the hotspots predicted by IUCN data are distributed in patches, and the hotspots are concentrated in the southwest of

Yunnan Province (Figure S3). The MaxEnt prediction results and IUCN prediction results were superimposed on the Pas, respectively, and it was found that their overlapping areas were also very different (Figure S4). If the MaxEnt prediction results are superposed with the hotspots predicted by IUCN data, the area of the protection hotspots of the existing PAs is 40,377.51 square kilometers, and the protection ratio is 61.50% (Figure S5, Table S7).

## 4. Discussion

### 4.1. The Existing Protection Efficiency Needs to Be Improved

　　The potential hotspot areas of species in this study are consistent with the hotspot results analyzed by other scholars using other methods. The differences are the results of this study found more hotspots in the Kunming–Qujing region than those predicted by Zhang et al. [39]. The formation of this hotspot is due to the strong influence of NDVI in the driving factors. Yang et al. (2016) identified more priority protected areas by combining animals and plants compared to the priority protected areas in Chuxiong [40]. This may be because Yang's team determines conservation priorities by county. This means that a small number of high priority areas within a district result in a county becoming a conservation priority. Our research results also have hotspots in Chuxiong, which is partly due to the differences in the methods adopted in the data analysis and visualization process. Yang et al. used the invest model, NPP (Net Primary Production) index, and topographic index to identify key areas for biodiversity conservation in 2021 [41]. The results of Yang et al.'s study shows that the key protected area in northwest Yunnan is wider. This may be due to the greater weight given to the topographic indicators, which affected the results. Therefore, although different research methods are used in this paper, the results are not much different from those of previous studies in the same research area.

　　The reason for the current low conservation efficiency may be because China's early PAs were specifically designed to conserve critically endangered species from extinction, so they lacked top-level design and systematic planning [65]. In addition, environmental disturbance and human activities threatening wildlife survival were also factors [66]. PAs in Yunnan Province play an important role in protecting forests and RESs. However, the specific needs of RESs in the region have not been fully considered. As can be seen from the distribution map of PAs in Yunnan Province, the distribution of all types of PAs in Yunnan Province is relatively even (Figure 1). However, the distribution of the most tightly managed and effectively protected nature reserves is not consistent with the distribution of RESs and protection hotspots. Our analysis shows that there are currently 156 various-level PAs in Yunnan Province, covering 28,118 km$^2$. The overlap area between provincial level natural reserves and HRAs is 1369.36 km$^2$. However, the protection rate of provincial level PAs for HRAs is only 5.46%, and the remaining 94.54% of hotspots are located outside of PAs without any protective measures.

　　The model shows us the potential range of RESs in future scenarios. The comparison shows that existing PAs are slightly better protected when low carbon emissions are maintained over long time scales. But when carbon emissions rise, current PAs will struggle to protect RESs. The prediction results of IUCN data and the MaxEnt model differ greatly in spatial distribution. The reason for this phenomenon is that the IUCN database does not match the actual situation in China [67–69]. After superimposing the predicted results of IUCN and MaxEnt models, the area is 136,382.34 km$^2$ (Figure S4), accounting for 34.61% of the total area of Yunnan Province, which is in line with the protection target of GBF. Therefore, combining the prediction results of IUCN data can protect more RESs and expand the coverage of the protection system more accurately.

　　By analyzing the coverage of species hotspots in each protected area, several PAs with high protection efficiency were found, such as Baima Snow Mountain Nature Reserve, Gaoligongshan Nature Reserve, Xishuangbanna Nature Reserve, etc. The majority of these areas are situated in Western Yunnan, consistent with previous research findings [70]. The conservation efficiency of the national nature reserve is higher compared to other PAs, which aligns with the findings of Wang et al. [71]. Located outside the three concentrated

distribution areas, the protection efficiency of the PAs in central Yunnan Province is low. Overall, national protected areas have higher conservation efficiency than other levels, which aligns with previous research findings [72].

*4.2. Climate Change Promotes Habitat Migration*

Most RESs are more sensitive to environmental changes than common species because of their small populations [73]. From the potential habitat transfer matrix, it can be seen that under the carbon emission level roughly consistent with the current emission pattern (RCP6.0), the area of large adaptation hotspots will continue to increase from the present to 2050 and 2070. The areas with a large abundance of RESs in Yunnan Province are concentrated in three regions, namely the western mountainous area of Yunnan, the Xishuangbanna–Wenshan high temperature area of Southern Yunnan, and the abundant vegetation area of Central and Eastern Yunnan. The distribution area of Western Yunnan is in agreement with the findings of Qi et al.'s study on *Cinnamomum mairei*, Yang et al.'s study on *Alsophila spinulosa*, and Liu et al.'s study on *Bhutanitis thaidina* [74–76]. The high temperature distribution area aligns with the research conducted by He et al.'s study on Asian elephants and Daniele et al.'s study on the fern, who studied RESs in Xishuangbanna's hotspots [77,78]. Wei et al.'s study about *Excentrodendron tonkinense* conducted a focused investigation on RESs in Wenshan, located in the southeastern part of Yunnan Province [79]. The high precipitation areas in Yunnan Province are distributed in Western Yunnan and at the low latitudes of the tropics [80]. The most dense vegetation cover areas in Yunnan are located near Gaoligong Mountain, Xishuangbanna, and Qujing in Western Yunnan. At the same time, the western mountainous region of Yunnan exhibits significant and abrupt variations in its topography. The distribution pattern of HRAs was most likely related to precipitation, vegetation cover, and topographic changes.

As shown in Figure 3, the increase of hotspots in the due west and due east directions is related to the expansion of hotspots in the Dehong, Baoshan, Kunming, and Qujing regions. This demonstrates that, in accordance with Zhang et al.'s findings, RESs tend to migrate towards mountainous regions. Furthermore, our analysis of topographic parameters supports this conclusion [81]. Simultaneously, the extents of the hotspots in Xishuangbanna have shrunk. But in the future, even with the relocation of locations, the results of the study still show that the high abundance of species will remain concentrated in three regions. This fits Penman's point according to the statistics of bioclimatic factors that affect species distribution [82]. The three factors that have great influence are the precipitation factor, the topographic factor, and the normalized vegetation index. This result is also consistent with many previous studies [44,83,84]. In particular, in Singh's study of India, which is adjacent to Yunnan, the most important factor affecting vegetation distribution is precipitation, which is consistent with our results [85]. This suggests that, as carbon emissions increase and the climate warms, species move to more suitable areas rather than becoming extinct.

The results of an evaluation of the land-use composition in hotspot regions are displayed in Table S8. The results indicate that in high richness hotspot areas, arable land and human activity on the land account for 29.01% (27,119.83 km$^2$) of potential hotspot areas. The primary threats to the potential habitat of RESs in Yunnan Province are the conversion of arable land and the encroachment of private forests. Yunnan has developed agriculture. Cash crops such as tea, rubber, and coffee are widely planted in hotspot regions. The expansion of these artificial forests and agricultural land has led to the fragmentation of RES habitats, leading to the degradation or complete loss of rich habitats for RESs [86–88].

Changes in the transfer matrix suggest that hotspots for RESs are not fixed but change with time and climate. In addition, the change in the richness degree of a region in a short period will not be too large. However, the change in the richness degree for RESs on a longer time scale will be large [89]. Except for the carbon emission model of RCP8.5, there is no transition from an HRA to an LRA in the period from 2050 to 2070 because the area of HRA is greatly reduced towards the two centers of east and west during this period,

and the centripetal shrinkage of MRA and LRA is also changed. Therefore, most UAs are transformed into MRAs and LRAs.

In particular, because plant populations cannot migrate, the spread is extremely limited [90]. This requires the creators and managers of PAs to pay attention to the protection of the potential habitats of RESs in the future during the planning phase.

We extracted the distribution area of HRA in each prefecture-level city in Yunnan and listed the top three cities with hotspot areas under each carbon emission scenario. Therefore, if the protection priority area is divided according to the administrative region, Baoshan, Honghe, and Dehong should be selected (Table S9). Both Kunming and Dali have large concentrations of hotspots, primarily due to their higher vegetation coverage. Moreover, Dianchi Lake and Erhai Lake serve as significant ecosystems for several avian species and aquatic organisms. Therefore, it is imperative to allocate additional focus to these two significant lakes.

### 4.3. Suggestions for Protection

The GBF, adopted at the fifteenth session of the Conference of the Parties to the United Nations Convention on Biological Diversity (COP 15), stipulates that at least 30% of the world's land and marine areas should be protected by 2030. Therefore, because of the poor effect of the current conservation system on the protection of RESs, this study puts forward the following suggestions.

(1) Fill in the existing protection gap. According to the conservation targets set by COP15, the current system of PAs is far from adequate for the protection of the RESs selected. At the same time, it is insufficient to protect key areas of ecosystem services [91]. This can be achieved by canceling the PAs of unprotected RESs where ecosystem services cannot be secured [68]. To solve this, managers need to flexibly set new PAs in the hotspot area of RESs distribution [92]. At the same time, the adjustment of PAs should also take into account the actual existence of species data. The governing body should strengthen the surveillance and investigation of RESs in their natural habitats. Relevant departments should pay attention to the existence and disappearance of rare and endangered wild species in reality, especially the migration patterns of rare and endangered animals. They should promptly modify the location of PAs [93].

(2) Protection and restoration of vegetation and control of the expansion of the scope of human activities. From our data, we can see that woodlands and grasslands occupy a very high proportion, regardless of the mode and year of emission. At the same time, the NDVI driving factor of the distribution of RESs also occupies a high position. We believe that forests are the most important habitat patches for RESs in Yunnan Province, so it is necessary to strengthen forest management to protect more RESs.

According to the result analysis of the land-use data, human activities also have a significant impact on hotspots. There are potential conflicts in the relationship between man and land in these areas. These conflicts will reduce the protection efficiency of existing PAs [94]. In order to achieve effective conservation, these areas should be allocated more management resources. Therefore, the government needs to combine development with protection. Therefore, this study recommends governments develop alternative conservation measures. Proactive strategies should be developed to prevent the outbreak of human–land conflicts, implement standardized management, and achieve a win–win situation for conservation and development [95]. Among them are effective district-based measures, participatory governance, and providing alternative livelihoods [96,97].

(3) Develop more effective conservation strategies for future climate change. The research model clearly indicates that hotspots with a high concentration of RESs will be greatly affected by future global climate change. And over time, the species listed can also change [66].

Often adjusting the boundaries of the protected-area system in response to these changes is difficult and impractical to manage. The government should adopt a flexible approach to adapt to future changes in the protected list and climate change without

frequently adjusting the boundaries of PAs, such as creating a protection model that combines tourism and management. Alternatively, the government might be able to promote the establishment of national parks [92]. National parks, as a distinct category of natural protected areas, effectively balance ecological preservation and recreational activities without compromising the level of protection provided by protected areas [98].

*4.4. Research Limitations and Future Prospects*

After research by many scholars, it has been confirmed that the MaxEnt model simulates species distribution [99,100]. Upon comparing our findings with the observed species distribution, we discovered that certain high altitude regions lack any instances of species branching. However, according to the model simulation results, these areas still hold potential for species distribution. This phenomenon is where the MaxEnt model deviates from reality. In further work, some cold regions that are indeed unsuitable for species can be excluded from the simulation results, improving the accuracy of predicting potential distribution areas.

One characteristic of the results of this study is that, in future scenario simulations, the range value of richness is significantly higher than the current predicted species richness. The emergence of this result is due to the characteristics of the bioclimate variable model selected for future scenarios. The BCC-CSM1-1 selected for prediction in this study is a climate-system model developed by the Beijing National Climate Center, which has played a good role in many studies [101,102]. However, compared to other system models, this climate-system model has multiple indicators with larger prediction results [103–105]. The error of a certain bioclimatic variable may have a decisive impact on the prediction results of this family or species group in the MaxEnt model. Due to the large number of families (species groups), the selected variable types were not adjusted based on each species, which is also the reason for this problem. In future research, our team will strive to improve this issue.

In addition, our analysis and research only conducted model simulations based on existing species presence data but did not consider the possibility of animal–plant interactions or animal interactions. Future research should also focus on the distribution of other species closely related to the research object. This prediction will provide a more scientific analysis of the existence areas and temporal changes of RESs and can also provide more accurate protection of biodiversity.

**5. Conclusions**

In situ conservation is a crucial part of biodiversity conservation, so it is particularly important to establish a precise conservation system. The MaxEnt model predicts the current and future distribution of RESs in Yunnan Province. The study revealed that the areas with the highest concentration of endangered species were primarily located in the northwest and east regions of Yunnan Province. In addition, they were located in the Xishuangbanna region. The places of "low richness" in Yunnan Province are primarily found in the central and northern regions. In the future climate model, the distribution location of hotspots does not change much, but the distribution area expands in the west direction. The predicted species distribution hotspots have a large area affected by humans and are also widely distributed in forest areas. Compared with the species distribution results simulated by IUCN data, the conservation efficiency of the current conservation system for RESs is low. This may be because the protection area has been established for a long time, and the protection objects and protection targets in the past are different from those in the present.

In view of these phenomena, this study suggests the following proposals. Relevant departments should combine the forecast results and pay attention to the existence and disappearance of rare and endangered wild species in reality. It is necessary to modify the protection strategy and address the current protection deficit. Previous studies have highlighted the potential for human–land conflicts due to the significant impact of human

activities on protected areas. Therefore, governments should develop alternative conservation measures in these areas, such as other effective district-based measures, participatory governance, and providing alternative livelihoods. Last but not least, governments need to develop effective conservation strategies for future climate change, adapting to future conservation list adjustments and climate changes.

**Supplementary Materials:** The following supporting information can be downloaded at: https://www.mdpi.com/article/10.3390/land13020240/s1, Figure S1: Changes in high-richness areas over time under different emission modes; Figure S2: Hotspot distribution predicted by IUCN data; Figure S3: Overlap hotspot distribution predicted by MaxEnt model and IUCN data; Figure S4: Distribution of protected hotspots in MaxEnt forecast and IUCN data forecast results; Figure S5: The IUCN hotspot area and MaxEnt hotspot area merged region; Table S1: The classification and classification criteria of rare and endangered species selected in this study; Table S2: The list of species used in the article is classified by the family; Table S3: Environmental variables used for modeling Yunnan province; Table S4: Correlation analysis of 19 biological environmental variables; Table S5: AUC value in the current and various situational model running processes; Table S6: Richness area transfer matrix under RCP4.5,RCP6.0 and RCP8.5 scenarios; Table S7: Data comparison of two prediction models; Table S8: The current proportion of land use types in hotspots; Table S9: Ranking of hot spot areas in prefecture-level cities.

**Author Contributions:** Conceptualization, Y.L., Y.B. and Z.H.; Methodology, Y.L., Y.B., Z.H. and J.W.; Software, Y.L. and Z.H.; Validation, Y.L., Y.B. and Z.H.; Formal Analysis, Y.L.; Investigation, Y.L. and Y.B.; Resources, Y.L., Y.B., J.W. and H.C.; Data Curation, Y.L.; Writing—Original Draft Preparation, Y.L.; Writing—Review & Editing, Y.L., Y.B., Z.H. and M.A.; Visualization, Y.L., Y.B. and Z.H.; Supervision, Y.B.; Project Administration, Y.B.; Funding Acquisition, Y.B. All authors have read and agreed to the published version of the manuscript.

**Funding:** This research was funded by the 14th Five-Year Plan of the Xishuangbanna Tropical Botanical Garden, Chinese Academy of Sciences (E3ZKFF7B), Yunnan Province Science and Technology Department (202203AP140007), Key Research Program of Frontier Sciences of Chinese Academy of Sciences (ZDBS-LY-7011), Trans-boundary cooperation on biodiversity research and conservation in Gaoligong Mountains (No. E1ZK251).

**Data Availability Statement:** The data presented in this study are available in Table S3.

**Acknowledgments:** The authors would like to thank the anonymous reviewers for their comments and suggestions.

**Conflicts of Interest:** The authors declare no conflicts of interest.

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
