# Peer review of "Spatio-Temporal Changes and Habitats of Rare and Endangered Species in Yunnan Province Based on MaxEnt Model"

_land, doi:10.3390/land13020240_

Round 1

Reviewer 1 Report

Comments and Suggestions for Authors

The paper simulates the distribution of rare and threatened species in Yunnan/China province under different carbon concentration scenarios as well as the extent to which these species are or will be under different forms of protection such as protected areas or National Parks. 

To improve this manuscript I recommend the following:

Avoid using abbreviations in the abstract;

Possible rephrasing of the statement about increased consumption among the middle class, only the middle class is increasing its consumption?

Change the colours used in Figure 1 from shades of grey to conventional colours for relief shapes similar to those used in Figure 4.

Redesigning the head of table 2 to make it easier to read and follow, possibly using abbreviations explained in the table heading.

NDVI abbreviation appears without explanation

On line 340 Yang et al 2016 to be written in capital letters

Lack of space between text and brackets or additional spellings without these being necessary Rows 359, 382, 389, 412, 414, 426, 430, 469, 496, 489, 501, 513, 487

Rephrase the statement in paragraphs 459-476 regarding relocation of protected areas which may be biased because the role of protected areas is not only to protect rare/vulnerable species but also to manifest ecosystem services.

Alignment of sub-chapter 4.4

Author Response

Thank you very much for taking the time to review this manuscript. Your comments are very important and meaningful, and these suggestions have helped us to improve our article. We express our sincere gratitude to the you for this invaluable opportunity to refine our work. Please find the detailed responses below and the corresponding revisions/corrections highlighted/in track changes in the re-submitted files. 

Because there are many contents, our point-to-point replies are attached in word, please download and check.

Reviewer 2 Report

Comments and Suggestions for Authors

Land-2813550 titled "Spatio-temporal changes and habitats of rare and endangered species in Yunnan province based on MaxEnt model

My comments on the manuscript are as follow:

1.     I have gone through the manuscript thoroughly, the paper is important and may of interest to a wider community. The authors have used used Maximum Entropy model to predict the present and future potential habitats of rare and endangered species in Yunnan Province.

2.     The MS is based on/dealing with rare and endangered species of Yunnan province. However, there are no details of both in the introduction section or how rare species will be defined/classified? Similarly, how many species were used/assessed in these stands/areas altogether, or what method was used for species collection/reporting. I could not locate these information and that needs to be explicitly mentioned/elaborated to help readers and future endeavors in the same field.

3.     I would specifically like to learn on how is this modelling approach (predicted here) is fitting into a real world scenarios particularly for identification of hotspots. As the same areas/allied regions have been studied earlier or at least information is available using other approaches. Such information will be very handy and may be provided in any relevant section.

4.     I could not see scientific justification particularly in the discussion section “why will under both current and future scenarios, rare and endangered species in Yunnan Province be primarily confined to the western mountainous region. When the authors themselves have referred to the fact that climate change will play a significant role in species migration/redistribution and precipitation levels will be playing a crucial role/key factor.

5.     It is understandable that CO2 has a determining role in shaping the global biodiversity. However, the authors have given more attention to one for few factors only. For instance, why so much focus is being placed on the levels CO2 alone, and nearly all other factors are left out (see abstract section).

6.     I also could not see the ethical approval of the study, who/which body approved this study if the plants contain collection from restricted/reserved parks or areas.

7.     I could not locate how identification of the specimen/species was done? Only reporting one database may or may not provide sufficient/enough information (see below also).

8.     L148: Although, the authors have mentioned that the flora identification/naming is based on https://www.catalogueoflife.org/. Plants valid/accepted names may be double checked and compared to the World Flora online database at: http://www.worldfloraonline.org/. Similarly, there must be a valid authority along the species name (at least once if and whenever mentioned).

9.     L29, 153 and elsewhere: All abbreviations needs to be described in full at their first place of mention e.g.  PAs, DEM and others elsewhere in the MS.

10.  L31-32: The recommendations/way forward given in abstract (as well as conclusion at the end see below) “….. Yunnan Province, with its rich species resources, has the potential to become a global innovator in biodiversity conservation by implementing improved conservation strategies” is very general and there are no details of how this is possible or could be achieved. Similarly, in conclusion section the recommendations are directed towards government and are too general and are not  specifically drawn/supported by the results of the study. I would suggest to enlist the current recommendations as well (if needed) but there must be very specific recommendations that are based on the current results.  

11.  L148-149: what is meant by species groups, this need clarity as what taxonomic unit is regarded as a group (is it above or below family level). Further, it is desirable that no other than internationally accepted/standard taxonomic groups/units may be used. “……..A total of 164 families and species groups, along with 3926 distribution points for representative rare and endangered species, were collected in this research …….”.

12.  L158-159: standard units needed to be mentioned along all quantitative data sets throughout the MS e.g. “……By 2100, the atmospheric carbon dioxide equivalent ……… exceeded 1370, 850, 650, and 490….”

13.  There are few minor issues and may be addressed:

a.     Abstract L18: The results are as follows may be written as “The results revealed that in both……”

b.     L25: …..“Furthermore, the analysis of conservation gaps also revealed significant conservation gaps”… may be written as …..“Furthermore, the analysis revealed significant conservation gaps…..”…

c.     L160: was may be replaced by were in “To avoid overfitting, a correlation analysis of 19 bioclimatic variables was carried out”.

d.     English language needs to be critically reviewed once more, there are typos and grammar related issues within the MS. Few examples of typos are given below whereas grammar needs stringent overview of the whole text. For instance “….Under future climate change, these areas will undergo significant changes that will effect the habitats of ecological protected areas”…. “affect” may be replaced by “affect” and the “habitats of ecological protected areas” may be replaced by “habitats and ecology of the protected areas”

e.     The intext references are not consistently mentioned. Also Ref. No. 3 and 14 along other may be overlooked for inconsistencies. Similarly, all intext references may be double checked with the list at the end and vice versa.

f.      L253-254: Figure legends here and elsewhere have very limited information and are not self-explanatory. Additional information may be added to legends. Further, key given for Figure 2 may be thoroughly looked again as the “study area” in key given cannot be depicted form the Figure.

Decision:

While the study is within the scope of the journal, and information may be handy and of wider interest and the MS may be accepted for publication after inclusion of the aforementioned suggestions”.

Comments on the Quality of English Language

See comments

Author Response

(The authors gave the same response as above.)

Reviewer 3 Report

Comments and Suggestions for Authors

The manuscript is a significant contribution in the field of habitat suitability modeling of

 Rare and endangered species using maxent along with other analytical tools like gap analysis etc. Authors have used the best possible combination of specific variables. I have some suggestions for the improvement of the manuscript.

Author Response

(The authors gave the same response as above.)

Reviewer 4 Report

Comments and Suggestions for Authors - p. 3: the article cites "variety of climates" without defining them.
Before talking about climate change or potential climate, we must at least
describe the current climate of Yunnan. - The article distinguishes national parks, nature reserves and natural parks.
It must add their Chinese translation (characters + pinyin) - Fig. 1: . Do “nature reserves” represent only “national nature reserves” or
all nature reserves, all administrative levels combined? . Qvjing = Qujing . the toponyms in white are difficult to read; it must add them in the
legend that they correspond to the prefecture-level cities - p. 7: Fig.S2 or Fig.2? - Generally speaking, the text is too long and quite confusing;
it is necessary to synthesize, reduce (max. 15 p.) and clarify.

Author Response

(The authors gave the same response as above.)

Round 2

Reviewer 4 Report

Comments and Suggestions for Authors   The article is still unnecessarily too long, but that's okay.